# Mast Cells in Upper and Lower Airway Diseases: Sentinels in the Front Line

**DOI:** 10.3390/ijms24119771

**Published:** 2023-06-05

**Authors:** Giovanni Costanzo, Giulia Anna Maria Luigia Costanzo, Lorenzo Del Moro, Emanuele Nappi, Corrado Pelaia, Francesca Puggioni, Giorgio Walter Canonica, Enrico Heffler, Giovanni Paoletti

**Affiliations:** 1Personalized Medicine, Asthma and Allergy, IRCCS Humanitas Research Hospital, 20089 Rozzano, Italy; giovanni.costanzo@humanitas.it (G.C.); lorenzo.delmoro@humanitas.it (L.D.M.); emanuele.nappi@humanitas.it (E.N.); francesca.puggioni@humanitas.it (F.P.); giorgio_walter.canonica@hunimed.eu (G.W.C.); giovanni.paoletti@hunimed.eu (G.P.); 2Department of Medical Sciences and Public Health, University of Cagliari, 09124 Monserrato, Italy; giuliacostanzo14@gmail.com; 3Department of Experimental and Clinical Medicine, University of Florence, 50121 Florence, Italy; 4Department of Health Sciences, University ‘Magna Græcia’ of Catanzaro, 88100 Catanzaro, Italy; pelaia.corrado@gmail.com; 5Department of Biomedical Sciences, Humanitas University, 20072 Pieve Emanuele, Italy

**Keywords:** mast cells, allergy, rhinitis, asthma, rhinosinusitis, nasal polyps

## Abstract

Mast cells (MCs) are fascinating cells of the innate immune system involved not only in allergic reaction but also in tissue homeostasis, response to infection, wound healing, protection against kidney injury, the effects of pollution and, in some circumstances, cancer. Indeed, exploring their role in respiratory allergic diseases would give us, perhaps, novel therapy targets. Based on this, there is currently a great demand for therapeutic regimens to enfeeble the damaging impact of MCs in these pathological conditions. Several strategies can accomplish this at different levels in response to MC activation, including targeting individual mediators released by MCs, blockade of receptors for MC-released compounds, inhibition of MC activation, limiting mast cell growth, or inducing mast cell apoptosis. The current work focuses on and summarizes the mast cells’ role in pathogenesis and as a personalized treatment target in allergic rhinitis and asthma; even these supposed treatments are still at the preclinical stage.

## 1. Introduction

Among all the unique cells that contribute to creating the complex and fascinating network which is the immune system, mast cells (MCs) are among the most mysterious and the least understood.

MCs are highly granular tissue-resident cells, part of the innate immune and neuroimmune systems and key participants in allergy, anaphylaxis, and various other diseases. Mast cells are scattered throughout the organism but are especially prominent close to barriers such as the skin and the mucosae of the lungs, digestive tract, conjunctiva, and nose [1,2].

In the army of the immune system’s cells, MCs may be considered the sentinels, the soldiers in the front line, deployed to the borders to sense the surrounding environment, able to rapidly respond to external stimuli and initiate a coordinated program of inflammation and repair. Indeed, although the main focus in understanding MCs has always been on their pathologic implications, mostly in allergic diseases, MCs also play an important and often underestimated role in tissue homeostasis, response to infection, wound healing, protection against kidney injury, the effects of envenomation and, in some circumstances, cancer progression [2,3,4]. Since providing a detailed picture of MCs’ biology goes beyond the scope of this review, this chapter will only provide an overview of MCs’ main features, with a focus on the characteristic which later will prove useful to understand the role of these cells in the pathophysiology of upper and lower airway diseases.

### 1.1. Origin and Development

MCs were first described by Paul Ehrlich in 1877 as aniline-positive cells around blood vessels in connective tissues [5] and were considered tissue-homed basophils for a long time [6], but more recent findings showed that the two cells develop from different hematopoietic lineages [7]. MCs are derived from CD34+ and CD117+ pluripotent hematopoietic stem cells [8] which arise in the bone marrow and develop through a granulocyte/monocyte lineage [9]. The most immature form of MCs detected to date has a hypogranular morphology and can be found in the yolk sac and fetal liver [10]. MCs are later released in the systemic circulation as undifferentiated mononuclear cells: both the release-in-circulation and the homing-to-tissue mechanisms are currently poorly understood (although many chemoattractants have been identified [11]).

Still, MCs increase in number and granularity over time and CD45+ CD117+ cells were observed in fetal skin and airways during the first weeks post-conception [12]. Moreover, human fetal skin contains IgE+ MCs as well, since it was proven that both passive and active prenatal sensitizations confers allergen sensitivity in utero [13].

In the adult, although most MCs inhabit peripheral tissue, immature human MC progenitors (MCps) can be found in peripheral blood as well as in the bone marrow as lymphocyte-sized cells with fewer granules than the mature counterpart [1,14].

KIT is well known as the most important signaling pathway for both maturation and survival of MCs, as further proved by the fact that imatinib treatment, which inhibits KIT signaling, depletes mast cells in vivo, although it spares circulating mast cell progenitors [15].

### 1.2. Subclasses

Although mature MCs are usually divided into subclasses based on their expression pattern of proteases (MCs prevalent in the skin producing both tryptase and chymase—MCTCs—and MCs that produce only tryptase—MCTs—more common in the airways), numerous studies over the past decade have already proven that there are countless MCs populations in different organs and within the same organ: this kind of heterogeneity in maturation, granule content, and expression in receptors elevates the risk of oversimplifying when classifying MCs [16]. The diversity of human MCs in the periphery is most likely the result of the interaction with different tissue-specific signals sent in response to various stimuli, such as infection, inflammation, and aging [1,17,18].

Nevertheless, the MCT/MCTC classification is still useful for several reasons: MCTs are cells at mucosal surfaces designed to interact predominantly with the immune system and regulate host defense [19], while MCTCs are proposed to be involved in tissue repair, fibrotic reactions, and angiogenesis [20].

The main differences between those specific phenotypes in vivo and how they may play a role in different diseases are yet to be revealed: the answers to those questions that have risen can be found presumably in a deeper understanding of MCps development and differentiation, as well as in the effect of the various activating and inhibitory signals on MCs [1].

### 1.3. Receptors

Their heterogeneity is not limited to protein expression, as MCs are the human cell type with more receptors than any other. Besides the well-known IgE receptors inducing allergies, MCs can adapt their repertoire of receptors by choosing from a wide and versatile range of non-IgE receptors [21] (e.g., FcγRI receptors that allow MCs to bind to IgG, tool-like receptors, C5a receptors, pattern recognition receptors, nuclear receptors, receptors for alarmins, integrins, neuropeptides such as substance P (SP), nerve growth factor (NGF), calcitonin gene-related peptide (CGRP) and vasoactive intestinal polypeptide (VIP), vitamin D): altogether, this variety of receptors allows MCs to adapt and meet the diverse functional requirements of their host tissue, and respond to a multitude of stimuli, such as proteases (tryptase), and complement bacterial products, neuropeptides, platelet-activating factor (also implicated in the pathogenesis of anaphylaxis [22]), hyperosmolarity and stress [23]. Moreover, it is worth mentioning that the recently described Mas-related G protein–coupled X2 (MRGPRX2) receptor, which is highly expressed in human cutaneous MCs (but absent in the airways) and can be activated by a vast number of cationic molecules such major basic protein and eosinophil cationic protein, neuropeptides, host defense peptides, and various drugs, causing pseudo-allergic reactions [2,21].

Remarkably, psychological stress in animal models induces MCs’ activation via the stimulation of peripheral nerves, and the release of neuropeptides and hormones, which may explain the hypothetical link between stress and asthma exacerbations [11,24,25]. A deep understanding of the subsequent biological cascade might become crucial for clinical practice as some receptors are proven to play a critical role in the pathogenesis of common diseases (e.g., EMR2 in vibratory urticarial [26]), with both activating and inhibitory functions. Learning how to interact with MCs’ receptors might serve as a target for future therapy in disease such as drug hypersensitivity, allergic asthma, and chronic urticaria.

### 1.4. Signal Transduction

Spleen tyrosine kinase (SYK) and Bruton’s tyrosine kinase (BTK) are two of the most studied and promising MCs signal transduction proteins: they both come next to the high-affinity IgE receptor (FcεRI) activation, which induces calcium release from stores in the endoplasmic reticulum, and then phosphorylate various targets which stimulate mediator synthesis and release pathways. BTK has been especially studied as a therapeutic target for MC-related diseases with promising data: irreversible BTK inhibitors have been shown to prevent IgE-mediated degranulation and cytokine production in MCs, block allergen-induced contraction of isolated human bronchi and, in a mouse model, prevent moderate IgE-mediated anaphylaxis and protect against death during severe anaphylaxis [27,28].

In addition to SYK and BTK, many other MC signal transductors have been demonstrated to have a therapeutic target potential, such as calcineurin [29], extracellular-signal-regulated kinase (ERK) [30], and hypoxia-inducible factor 1-α (HIF-1α) [1,31].

### 1.5. Granules

On the structural level, MCs are granulocyte usually stained with toluidine blue (but also May–Grunwald–Giemsa is effective in identifying mast cells from nasal mucosa samples such as nasal cytology specimens) whose cytoplasmatic granules contain mainly histamine and heparin: exocytosis by degranulation is mostly, but not exclusively, triggered by the binding between IgE’s specific antigen and IgE, abundantly attached to the surface of MCs thanks to the presence of the high-affinity receptor (FcεRI) for the Fc region of IgE [1].

However, in many diseased tissues, including the asthmatic bronchial mucosa, MCs typically present piecemeal degranulation, an unconventional secretory pathway characterized by vesicular transport of small packets of materials from cytoplasmic secretory granules to the cell surface [32]. The main difference between piecemeal and anaphylactic degranulation is that the first is characterized by the presence of variable losses of dense contents from granules, while the traditional or anaphylactic degranulation consists in the extrusion of membrane-free granules into newly formed degranulation channels in the cytoplasm or through pores in the plasma membrane connected to the exterior environment, with the subsequent release of mediators into the extracellular space. The mechanisms leading to piecemeal degranulation in MCs are poorly understood [33].

It is important to highlight that anaphylactic degranulation or “compound exocytosis” is not a cytotoxic event for MCs, and even after almost complete degranulation human lung MCs are able to re-granulate over a period of 48 h [34].

### 1.6. Mediators

Once the proteins are released from the acidic environment of the granules, the neutral pH of the extracellular space generates active mediators. Histamine, the main mediator, leads to post-capillary venules’ dilatation, endothelium activation, and increased permeability of blood vessels which, altogether, result in the cardinal signs of inflammation (local edema, warmth, redness, heat). These reactions occur wherever MCs degranulate, resulting in very different signs and symptoms in various diseases, from bumps, typical of urticaria, to nasal obstruction due to mucosal edema in allergic rhinitis. Nevertheless, MCs can adapt their response and function depending on the stimulus and exhibit a full-blown response leading to degranulation and release of both pre-stored and newly synthesized mediators. Histamine is par excellence the main mediator associated with MCs, but tryptase is the most used as a serum biomarker, as a high blood level of this enzyme implicates the cells’ degranulation: increased chronic tryptase levels may be indicative of mastocytosis, a clonal MC disorder or hereditary α-tryptasemia, while acute elevation is associated with systemic hypersensitivity or anaphylaxis [1,34,35,36,37,38]. 

Moreover, the chronic release of MCs’ tissue-repairing cytokines eventually leads to deleterious effects for the host, such as fibrosis in asthma [11,39]. Overall, there is a vast number of different mediators: the most important ones will be discussed later in this article, with a specific interest in how different proteins contribute in their own way to the physiopathology and the clinical profile of various diseases. 

### 1.7. Interactions with Other Cells

Besides playing the role of the sentinels spread in connective tissues all throughout the organism, MCs also interact and cooperate with a multitude of other different cells, as the number and heterogeneity of MCs’ receptors might have already suggested. Eosinophils have always been considered the closest partner of MCs [40], but recent studies have enlightened important interactions also with T and B cells.

It is now demonstrated that MCs can induce T-cell activation in many ways: by direct antigen presentation [41], transferring antigens to professional antigen-presenting cells that will eventually interact with T-cells [42], shuttling antigens via exosomes [43,44] or priming dendritic cells, inducing the release of INF-γ and IL-17 that will eventually lead to Th1 and Th17 responses [45].

Less is known regarding the crosstalk between MCs and B cells, although it is proven that MCs can induce class-switch recombination in B cells [46], playing a role in antibody-mediated immunity.

Nevertheless, the most important partnership remains the one with eosinophils: MCs and eosinophils are the main components of the so-called “allergic effector unit”, defined by the cross-talk between these two cells mediated by both physical cell-to-cell contacts and the release of mediators, e.g., specific granular mediators, arachidonic acid metabolites, cytokines, and chemokines. The synergy between MCs and eosinophils crucially contributes to both allergic and non-allergic inflammation, determining the activation of both cell types and the consequent release of proinflammatory and chemotactic mediators, eventually leading to clinical effects that go beyond the sum of the two parts [40].

Overall, MCs are able to interact with virtually any other immune cell. Moreover, MCs are shown to interact also with nonimmune cells such as endothelial cells, neurons, fibroblasts, adipocytes, keratinocytes, osteoblasts, osteoclasts, and cardiomyocytes, although the details and possible implications of these relationships are yet to be fully understood [36].

### 1.8. Host Defense

Although commonly associated with hypersensitivity reactions, MCs are also capable of actively contributing to host defense and reacting to pathogens’ attacks both through direct responses (phagocytosis, extracellular traps, and cytonemes) and indirect responses (degranulation, which alters the infection environment, and with recruitment and activation of neutrophils, T cells, natural killer cells, eosinophils, and macrophages) [1,36]. In mice models, MCs are known to play a role in various bacterial infections, such as *Coxiella burnetii*, *Pseudomonas aeruginosa*, and *Helicobacter pylori*, among others. Moreover, MCs provide an important contribution during fungal and parasitic infections, e.g., *Plasmodium* spp., *Leishmania* spp., *Toxoplasma gondii*, and *Trypanosoma* spp. [47], and were long considered to be one of the cells mainly responsible for primary defense against parasite infection [48].

Finally, MCs are shown to be able to contrast viral infections by responding to either viral RNA or DNA recognition via TLR-3 and TLR-9, respectively [49].

In conclusion, MCs should be considered a double-edged sword; while they have the potential to contribute to host defense against infection, and manage cancer and tissue repair, the release of their potent mediators and cytokines can also mediate or aggravate pathologies; for example, causing a decline in airway function in individuals with lung pathologies [11].

### 1.9. Immunometabolism in Mast Cells

It has been known for almost a century that actively proliferating cells, such as cancer cells, arrest the Krebs cycle and begin producing energy mainly by aerobic glycolysis [50]. A few decades later, it was discovered that this metabolic switch is also performed by innate and adaptive immune cells [51,52,53].

Immunometabolism has recently become a research subject in pursuit of potential treatment for immunological and allergy illnesses [54].

The immune metabolism of the cells featured in this review is of relevance in allergic diseases; indeed, there is some evidence that aerobic glycolysis is required for the acute activation of MCs:

Chakravarty suggested that the utilization of glucose through glycolysis promotes the release of histamine from MCs [55]. In addition, the inhibition of MCs glycolysis was shown to attenuate IL-33-mediated MCs function and cytokine production in vivo [56]. Moreover, the glycolytic intermediates were found to be required for MCs’ degranulation in vitro [57].

This evidence suggests a link between MCs’ metabolism and effector function; indeed, immune metabolic analyses have recently identified glycolytic ATP production in MCs as potential targets to modulate allergic responses [56,58].

We believe it could be a promising and fascinating field of research.

## 2. Mast Cells in the Airway

MCs are scattered all over the body; in the respiratory system, the highest concentration is found in the trachea and large bronchi, mostly underneath the epithelium rather than within it. In contrast, the lungs do not present many MC progenitors in a physiological state, but they may be recruited in response to antigen-induced inflammation. Within the various compartments of the lungs there are distinct MCs subclasses, containing granules with different contents: the MCT subtype predominates in the lung parenchyma, bronchial *lamina propria*, and bronchial epithelium, while the MCTC subtype surrounds respiratory blood vessels, close to the endothelial cells. The presence of MCs within the nasal mucosa, bronchial epithelium, and respiratory blood vessel explains how these cells may provoke a plethora of symptoms in various respiratory diseases which will be discussed later on, such as increased mucous production, local edema, increased vascular permeability, and obstructed airflow [59].

Overall, MCs play both protective and pathological roles: they contribute to tissue repair, deactivation of proinflammatory cytokines, lympho-angiogenesis, smooth muscle relaxation, and negative feedback with immunomodulant functions; on the other hand, MCs provoke muscle contraction, immune cell infiltration, airway barrier disruption, airway remodeling, fibrosis, and increase mucus production [60]. In this article, the focus has been put on the two main MC-related airway diseases, respectively from upper and lower airways: rhinitis (with a particular interest in allergic rhinitis) and asthma. As for most inflammatory diseases, the precise mechanisms which lead to MCs’ dysfunction in asthma and rhinitis are yet to be understood: a deeper understanding of the starting pathways that eventually lead to MCs’ dysfunction will contribute to develop therapeutical strategies able to intervene upstream in those airway diseases’ pathogenesis.

### 2.1. Upper Airways: Allergic Rhinitis

In the heterogeneous group of rhinitis (defined as an inflammatory disease of the nasal mucosa), different subtypes can be identified based on etiology: infectious, inflammatory, medicamentous, hormonal, occupational, atrophic, and vasomotor; among the latter, characterized by nasal hyper-reactivity, two big subtypes can be distinguished: allergic rhinitis (AR) and non-allergic rhinitis (NAR) [59].

Allergic rhinitis (AR) is a common chronic IgE-based inflammatory rhinopathy that affects up to 20–30% of adults and up to 40% of children worldwide. Typical symptoms include nasal congestion, anterior and/or posterior rhinorrhea, sneezing, and itching, but may include fatigue, attention, learning and memory deficits, and even depression as a result of sleep-disordered breathing and obstructive sleep apnea. When ocular symptoms are included, typically conjunctivitis, the disease is called rhinoconjunctivitis (ARC). Symptoms may be present year-round or seasonally, depending on the timing of allergen exposures [61]. AR is also consistently associated with asthma (some studies report that about 78% of patients with asthma also had AR [62]).

Similarly to other allergic diseases, AR is the result of an inordinate response of the immune system of a sensitized individual to some innocuous triggers, known as allergens. Allergens are usually divided according to temporal patterns during the year, as either perennial (caused by mold, dust mites, house pets, occupational allergens) or seasonal triggers (caused by pollens and molds, which vary greatly depending on the geographic area). The presence (or absence) of seasonal symptoms allows one to discriminate between seasonal allergic rhinitis, caused by seasonal peaks in the airborne load of pollens, from perennial allergic rhinitis, caused by allergens present throughout the year [63].

The diagnosis is clinical, suspected when the patient experiences one or more of the hallmark symptoms in response to allergen exposure. Some other typical examination findings, especially common in children, are Dennie-Morgan lines (increased folds below the lower eyelid) and so-called allergic facies, with high arched palate, mouth breathing, and dental malocclusion. Determining specific IgE positivity may be helpful to confirm the diagnosis, and identifying the trigger is crucial to develop an effective therapeutical strategy, such as in guiding avoidance measures or immunotherapy. Skin testing is the safest, least invasive, most used test, and provides rapid results with an excellent sensitivity and good specificity, whereas blood-specific IgE testing is more specific but with less sensibility; overall, the two tests are complementary [61]. In selected cases, the causal relation between exposure to a specific antigen and the development of symptoms can be assessed by nasal allergen challenge.

A sino-nasal disease associated with AR is rhinosinusitis, a rhinopathy in which the inflammation also involves paranasal sinuses, characterized by specific symptoms such as facial pressure and loss of the ability to smell. Chronic rhino sinusitis (CSR) is instead defined as the persistence of typical symptoms for more than 12 weeks and can be phenotypically divided in CRS with nasal polyps (CRSwNP) and without nasal polyps (CRSsNP). The difference between the two forms goes beyond the phenotype, as it is now proven that most CRSwNP patients present a type 2 inflammation endotype, with a nasal mucosa rich in eosinophil and MCs [64], as will be discussed later.

#### 2.1.1. MCs in the Pathogenesis of AR

MCs are proven to play a key role in both initiating and maintaining the main clinical features of AR (Figure 1).

In nasal mucosa, antigen-presenting cells process inhaled allergens and present them to naïve CD4+ T lymphocytes in the draining lymph nodes, where they are activated and transformed into T-helper 2 CD4+ lymphocytes under IL-4 stimulation. Th2 cells, with type 2 innate lymphoid cells, release, among other cytokines, IL-4 and IL-13, which altogether stimulate specific B-cell lymphocytes to differentiate into antibody-producing plasma cells. These IgE antibodies adhere on MCs’ surface, making them ready to respond to the same inhaled allergen which initiated this process in the first place [63].

When triggered by the allergen, MCs secern via degranulation various cytokines whose effects describe and explain every main clinical feature of AR. Indeed, every AR hallmark symptom can be in some way traced back to MCs’ activation. 

In the nasal mucosa, histamine induces activation of H1 receptors on sensory nerves of the afferent trigeminal system (causing itching and sneezing) and parasympathetic nerves (leading to mucous gland stimulation and, therefore, rhinorrhea). The activation of H1 and H2 receptors in nasal blood vessels contributes to increased vascular permeability and vasodilatation in the nasal mucosa, causing nasal congestion [37]. Local edema and enhanced inflammatory cell infiltration are also exacerbated by MC-released growth factors (e.g., fibroblast growth factor 2, FGF-2, and vascular endothelial growth factor, VEGF-2), which also contribute to vasodilatation and vascular permeability [59]. MCs prolong inflammation and its symptoms as well, with the release of late-phase response cytokines and chemokines: prostaglandin D2 (PGD2) and cysteinyl leukotrienes also increase vascular permeability, promote the recruitment of type 2 innate lymphoid cells and, finally, eosinophil, which ultimately lead to the damage of nasal epithelium.

Usually, the late phase coincides with the resolution of the inflammatory reaction, but continuous exposures to allergens may lead to a chronic phase in which the local allergic reaction fails to resolve [37].

#### 2.1.2. Therapeutical Perspective on MCs’ Role in AR

Therapy for AR is well established and based on three complementary cornerstones: allergen avoidance (also with high-efficiency particulate air filters, barrier measures such as non-pharmacological intranasal cellulose powder and lipid microemulsion [65]), pharmacotherapeutic agents (which generally treat only symptoms and do not address the underlying cause of the allergic inflammation), and immunotherapy (IT). Usually, this three-headed approach is effective and through the analysis of this strategy and how it acts on MCs, it is easy to understand the key role they play in AR. Actually, most of the available therapeutic and symptom-control strategies in allergic diseases, including AR, aim largely at targeting MCs to suppress the allergic process [61].

Indeed, MCs’ main product is histamine and the most important first line of treatment for AR is antihistamines. Antihistamines are now divided into first and second generation: whereas the latter tend less to cross the blood–brain barrier inducing sedation, both are effective at controlling symptoms of AR, blocking histamine receptors expressed on the nasal mucosa and preventing their activation by MC-secreted histamine. This result alone is effective in reducing symptoms of rhinorrhea, sneezing, and nasal itching [66].

Currently, the term antihistamine refers to anti-H1 drugs; however, both anti-H3 and anti-H4 antihistamines are under development and are already proven, respectively, to modulate the vascular contractile response in the human nasal mucosa and to act as anti-inflammatory molecules. None of these drugs are approved to date [65].

Finally, intranasal antihistamines (e.g., azelastine) are also available and are equal to or more efficacious and rapid in their symptomatic relief than oral antihistamines, with minimum adverse effect [63].

Another possible first-line symptomatic therapy is intranasal glucocorticoids (GCSs), used alone or in association with both local and systemic antihistamines. GCSs, in general, suppress various stages of the allergic inflammatory reaction as a result of modifications in gene transcription, leading to a reduction in the synthesis of cytokines, which ultimately attenuates the recruitment, survival, and activity of inflammatory immune cells, like MCs [59].

Other medications are specifically directed to modulate MC activity: the so-called MC stabilizers, such as cromolyn sodium (CS), prevent the degranulation and the subsequent release of inflammatory mediators. This is an adjunctive treatment in AR and is effective in reducing major symptoms; it also works as an immunomodulant, contributing to the attenuation of allergic inflammation [67].

Leukotriene receptor antagonists (LTRAs), such as montelukast and zafirlukast, block the binding between leukotrienes secreted by MCs and eosinophils, among others, and leukotriene-receptors on immune cells. Overall, anti-leukotrienes, in combination with antihistamines, are proven to be effective against AR and result in significant improvements in daytime nasal symptoms and quality of life [68].

Omalizumab is a human monoclonal antibody designed to block IgE and, among other targets, it minimizes MCs degranulation and the subsequent release of cytokines. In particular, omalizumab is a humanized antibody that binds to the Fc portion of human IgE and interferes in the binding of IgE to high-affinity receptors (FceRI) on effector cells including mast cells and basophils. Omalizumab is proven to reduce the Daily Nasal Symptom Severity Score (DNSSS) and improve the Daily Ocular Symptom Severity Score (DOSSS), and results of the Rhino-conjunctivitis Quality of Life Questionnaire. Overall, it reduces the consumption of antihistamines without exposing patients to any major adverse event [69]. Omalizumab delivered good results as an add-on therapy in patients with both asthma and AR, delivering a significant improvement of allergic rhinitis symptoms, in the asthma control test, and in lung function [70]. Omalizumab significantly improved endoscopic, clinical, and patient-reported outcomes in severe CRSwNP with inadequate response to intranasal corticosteroids. Furthermore, omalizumab was effective in severe CRSwNP, significantly improving endoscopic, clinical, and patient-reported outcomes [71].

Dupilumab is a fully-humanized monoclonal antibody designed to target IL-4Rα and IL-13 receptors: it inhibits IL-4 and IL-13 cascade and indirectly regulates MC activation. Dupilumab was found to improve clinical symptoms in a patient with asthma and perennial AR [72]. Moreover, dupilumab reduced polyp size, sinus opacification, and symptom severity in patients with CRSwNP [73], performing better than omalizumab at week 24 in an indirect treatment comparison [74].

Finally, the third main kind of treatment available for AR is allergen immunotherapy (AIT). AIT consists in scheduled administration of precises doses of allergen extracts with the goal of suppressing the underlying inflammatory diathesis and inducing a state of clinical tolerance to the relevant allergen, eventually reducing, if not arresting, the inflammation that underlines AR’s pathogenesis [63]. AIT is proven to suppress both innate and adaptive immune responses and works on the immune system via early desensitization of MCs and basophils, generation of regulatory T and B cell response, regulation of IgE and IgG4 production, and decreasing the overall activity of eosinophils and MCs in the mucosa [75].

### 2.2. Other Rhinopathies: The Role of Nasal Cytology

Recent data suggest that MCs act as innate immune cells against pathogens and initiate defensive responses within the epithelium of the respiratory tract; in fact, initial stages of viral rhinitis are characterized by the high presence of MCs which, among other immune cells such as T and B lymphocytes, start the inflammatory cascade in upper airways [76].

In fact, although the literature has always focused on the role of MCs almost exclusively in AR, recent studies have proven that the impact of MCs in pathogenesis on NAR has always been underestimated. Recent nasal cytology findings have demonstrated that MCs are critical cells in the physiopathology of several other rhinopathies, such as NARMA (NAR with MCs), NARESMA (NAR with eosinophils and MCs), and the aforementioned CRSwNP, with possible interesting effects on the therapeutical management of these diseases. This evidence expands the pathophysiological role of MCs in general over the boundaries of IgE-mediated diseases, although these aspects are still mostly yet to be studied [59].

Among NARs, NARMA is characterized by the presence in nasal cytology of MCs > 10% in the nasal mucosa, partially degranulated. This pathologic feature has repercussions in management since the clinical presentation is usually severe and associated with the presence of asthma with or without nasal polyposis. NARMA is now considered a transitional form of NARESMA, a more recently defined disease subtype characterized by the presence of eosinophils and MCs in variable proportion with relevant degranulation. In this scenario as well, the management is difficult as the inflammatory infiltrate is mixed and leads to a more severe disease, often associated with nasal polyposis, asthma, and rhinosinusitis [77].

Furthermore, MCs also have been recently shown to play a role in CRSwNP, a disease in which eosinophilic inflammation has always been considered the cardinal feature. MCs are highly present in nasal polyps; specifically, MCT expressing carboxypeptidase A3 and MCTC, which infiltrate the airways’ smooth muscle and subepithelial glandular tissues. These subtypes contribute to activate eosinophils, notoriously abundant in the disease [78]. Moreover, MCs’ concentration is higher in eosinophil-rich nasal polyps compared to the non-eosinophilic ones, suggesting the close relationship between these two cells in the pathogenesis of the disease [79]. Lastly, MCs have been proven to have a crucial role in the most severe form that is refractory to traditional CRSwNP treatments [59].

### 2.3. Lower Airways: Asthma

Asthma is a chronic respiratory disease affecting about 1–18% of the population in different countries, characterized by variable symptoms (wheezing, shortness of breath, chest tightness with or without cough) which can vary over time in intensity, from a temporary absence to episodic exacerbations that may be life-threatening for the patient.

Three of the most characteristic clinical features of asthma are:Variable expiratory airflow limitation, which puts asthma in the group of obstructive respiratory diseases. The major processes that determine this trait are airway inflammation, contraction of airway smooth muscle (ASM), excessive mucus secretion, and mucosal edema due to increased vascular permeability [80].Airway/bronchial hyperresponsiveness, maybe the most distinctive trait [81].In patients with long-standing asthma, persistent or incompletely reversible airflow limitation due to reduced respiratory muscle function, loss of elastic recoil, and airway wall remodeling. The key elements of this remodeling are denudation of the epithelial layer, goblet cell and mucous gland hyperplasia and hypertrophy, subepithelial fibrosis, abnormal extracellular matrix deposition, vascular proliferation, and increased ASM mass [82].

MCs, traditionally considered simple effectors, are proven to play a role in each one of these clinical features, and MCs’ impact on pathophysiology may vary from modest to predominant. In this article, a clinical-centered perspective has been favored, starting from the hallmark symptoms of asthma and then going backward to find the responsible molecule (Table 1) (Figure 2).

#### 2.3.1. Expiratory Airflow Limitation

Although also present in healthy airways, MCs typically infiltrate three districts in the asthmatic airways: ASM bundles, the epithelium, and the mucosal gland [11].

The key role played by MCs in the pathophysiology of expiratory airflow limitation and the interaction with ASM can be interestingly deduced by comparison with a disease that is similar, yet different in etiopathogenesis: eosinophilic bronchitis (EB), a corticosteroid-responsive respiratory condition characterized by cough and presence of sputum eosinophilia in absence of airflow obstruction and bronchial hyperresponsiveness. Although these two diseases share many similarities in immunopathology (e.g., the infiltration of T-cell cells, eosinophils and mucosal MCs, the pattern of membrane collagen deposition, the epithelial integrity, the key role of IL-4 and IL-5), they profoundly differ in two key features: abnormal airway smooth muscle (ASM) function characterizes asthma as an obstructive pulmonary disease, whereas in EB pulmonary function tests are normal. From a histopathological point of view, asthmatic patients’ ASM is abundant in MCs, whereas there are virtually none in subjects with EB or healthy individuals [83,84]. Along with the fact that in the ASM asthmatic biopsy there are almost no T cells or eosinophils, overall this evidence may suggest that MCs are one of the critical determinants of the asthmatic phenotype [11].

As a matter of fact, asthmatic ASM secretes many chemokines and growth factors that are proven to have chemotactic activity for MCs [85] (mainly CXCL10, which in fact is increased in subjects with asthma compared with normal subjects) [86]. Once present, MCs can interact with ASM both via localized mediator release and direct cell-to-cell contact, which ultimately leads to hypertrophy, hyperplasia, and airflow obstruction.

Indeed, MCs secrete through degranulation a multitude of mediators in ASM which, altogether, lead to some of the most prominent clinical features of asthma, such as shortness of breath, wheeze, and chest tightness: histamine induces bronchoconstriction and mucosal edema; prostaglandin (PG) D2 and leukotriene (LT) C4 are potent agonists for airway smooth muscle contraction and mucus secretion; tryptase induces bronchoconstriction in animal models; all the range of proinflammatory cytokines (e.g., IL-4, IL-5, and IL-13) regulate IgE synthesis, therefore playing a role in the etiopathogenesis itself; transforming growth factor β (TGF-β) and basic fibroblast growth factor (bFGF) act as profibrogenic cytokines, eventually leading to airway remodeling, a late and severe complication typical of asthma. Chymase’s role in less understood, but it is believed to inhibit T cells adhesion to ASM (which can explain the paucity of T cells within the muscle) and initiate the formation of collagen fibrils [11,38,39,87,88,89].

There are several pieces of evidence suggesting that the source of these mediators are actually MCs, such as the higher concentration of the preformed MC-specific protease tryptase in BAL after the bronchial allergen challenge [90] and the fact that salbutamol (which in vitro inhibits MCs degranulation) eliminates any asthmatic reaction and increases in histamine level [91].

Moreover, MCs seem to interfere with the physiological bronchodilatation that follows a deep breath and avoids airway narrowing, a protective mechanism that in asthma is impaired or lost altogether [92].

In conclusion, MCs, with their micro-localization within airway smooth muscle, may be the key cytologic features that set asthma apart from any other similar disease, such as EB [38]. Furthermore, this relationship might be even stronger in cases of asthma death, where the number of degranulated MCs within the ASM is higher in fatal compared to non-fatal asthma [93].

Nonetheless, the effect of MCs on the pathophysiology of asthma is not limited to the interaction with ASM. MCs also infiltrate the bronchial epithelium, a location consistent with their role as sentinels against noxious stimuli and aeroallergens. Many MCs mediators (PGD2, leukotriene C4 LTC4, IL-6, IL-13, TNF-α, tryptase, and chymase) interact with the epithelium, leading, altogether, to mucus hypersecretion by hyperplastic submucosal glands and epithelial goblet cells, which is a known key feature, especially in severe asthma [35,94].

A further molecule of particular interest is mast cell-derived amphiregulin, which contributes to epithelial goblet cell metaplasia and mucus hypersecretion in asthma, effects remarkably not suppressed by dexamethasone. In addition, it induces the proliferation of human airway fibroblasts, suggesting that it has a key role in subepithelial fibrosis, as will be discussed later [95]. Mucus hypersecretion by hyperplastic submucosal glands and epithelial goblet cells is an established feature of asthma in general, but it is especially relevant in severe and fatal asthma [96].

#### 2.3.2. Bronchial Hyperresponsiveness

MCs’ position in the superficial interface between the immune system and the environment makes them the ideal cells to rapidly respond to various external noxious stimuli and allergens, and the more plausibly responsible for bronchial hyperresponsiveness (BHR).

A dysfunctional pro-inflammatory airway epithelium, typical of asthma, contributes to MCs’ activation and enhances expiratory airflow occlusion. One of the mechanisms underlying MC recruitment, survival, and activation might be the expression of IL-13, which triggers stem cell factor (SCF) production by airway epithelium, as both IL-13 and SCF are elevated in asthmatics in comparison with healthy controls [97,98].

Another factor related to a more contractile phenotype of ASM is TGF-β, released by both MCs and ASM cells and correlated with the intensity of α smooth muscle actin (α-SMA), a protein that leads to increased spontaneous and provoked ASM contraction [99].

Among all the MCs present in ASM, some are more implicated with hyperresponsiveness, which are the ones that express fibroblast markers: the number of these so-called fibroblastoid MCs correlates closely with the severity of airway hyperresponsiveness as fibroblastoid MCs showed increased chymase expression and activation with an exaggerated constitutive histamine release [100].

Finally, the strategic role of TNF-α in asthma pathogenesis is well known: its immunoreactivity is increased in the airways of patients with mild asthma, its expression is increased markedly in patients with severe asthma, and it is shown to induce and exacerbate bronchial hyperresponsiveness. By using two monoclonal antibodies directed to different epitopes of IL-4, a 1994 study provided the first evidence for enhanced IL-4 secretion in asthma and demonstrated an increase in the number of MCs staining for TNF-α in the asthmatic biopsies, suggesting that, in this context, this cytokine is mainly produced by MCs [101]. Another later small double-blind placebo-controlled study on the use of etanercept (a biologic medical product that interferes with TNF pathways) in severe refractory asthma confirmed that the therapy led to improvements in quality of life, lung function, and bronchial hyperresponsiveness; the authors did not find a significant change in sputum inflammatory cells, but reported a marked reduction in sputum histamine concentration: this not only suggests that etanercept inhibited primary mast cell activation, but also that MC-derived TNF-α plays a critical role in asthma pathophysiology [102].

#### 2.3.3. Airway Remodeling

The third clinical feature of asthma is also the most challenging to treat: airway remodeling, consisting of deposition of type III and type IV collagen in the *lamina reticularis* below the true epithelial basement membrane [103]. The more probable origin for this collagen is proliferating myofibroblasts and, nowadays, many studies have demonstrated that also the interaction between MCs and fibroblast plays a role. The following are some of the cytokines secreted by MCs that are proven to interact with fibroblasts: histamine, basic FGF, TNF-α, and IL-4 induce fibroblast proliferation; IL-4 is also a fibroblast chemoattractant and directly stimulates collagen secretion; chymase and tryptase induce proliferation, chemotaxis, collagen synthesis, and fibroblast migration [88]; lung MC-derived TGF-β1 has the potential to induce fibroblast differentiation into contractile myofibroblast, while amphiregulin, as already mentioned, induces fibroblast proliferation in human airways [11].

Nonetheless, the central role in airway remodeling is played by growth factors, since the same signals that stimulate tissue repair and vascularization, if not regulated and later suppressed, eventually lead to changes in cell distribution, thickness, and stiffness in the respiratory tract. MCs’ heparin-binding epidermal growth factor (HB-EGF) promotes fibroblast proliferation and migration [104], whereas VEGF-A is a neoangiogenic factor correlated to a more severe form of asthma, as it induces mucus production and possibly provides additional vascular pathways to immune cells to infiltrate bronchial tissues [105,106]. Some other proteins contribute indirectly to airway remodeling increasing the expression of growth factors: e.g., LTC4 secreted by MCs activates the mitogen-activated protein kinase (MAPK) pathway and increases the expression of growth factors [107].

Moreover, serotonin, a neurotransmitter commonly known in neurology, has been shown to be elevated in patients with asthma, as it is one of the constituents of the MC granules released by IgE-mediated degranulation. According to studies of human cell culture, it is believed that serotonin may contribute to structural changes by promoting airway and vascular remodeling, and bronchoconstriction [108].

#### 2.3.4. Other Features

Neovascularization is a key feature in both physiological and pathological settings and has particular relevance in tumor growth. Nevertheless, vascular remodeling is also a characteristic of asthma and it is well associated with the presence of airflow obstruction [109]. The evidence of MCs adjacent to blood vessels and the expression by MCs of some of the most important angiogenic factors (basic FGF and, VEGF, IL-6, and CXCL8) suggest that they may have a role in airway neovascularization in asthma [60].

MCs are also associated with the cough reflex as MCs are in close proximity with nerve fibers within the lung and produce histamine and PDG2, well-known pro-tussive molecules and already proven to be increased in the sputum of subjects with eosinophilic bronchitis compared with healthy controls [110].

#### 2.3.5. Chronic Activation

So far, we have discussed the degranulation and the release of MCs cytokines, but we also must question which mechanism, among all the aforementioned, determines an ongoing chronic activation. Allergens are often referred to as mainly responsible for this, but allergen avoidance usually has minor effects on established asthma (especially in the occupational subtype), which appears to become self-perpetuating. Anti-IgE therapy markedly reduces both airway inflammation and mast cell activation, as manifested by reduced IL-4 expression, but has a minimal effect on BHR: this suggests that other factors are causally linked to airway hyperresponsiveness to methacholine in asthma [111]. This role is sometimes attributed to TLR3, the ligand for which is double-stranded viral RNA: influenza viruses and respiratory syncytial viruses are common causes of asthma exacerbations [112], but a substantial contribution might be given by G-protein-coupled receptors which respond to a wide range of stimuli, leading to an increase in some cytokines such adenosine A2B (known to determinate bronchospasm) [113].

In conclusion, the mechanism of MCs’ chronic activation in the asthmatic bronchial mucosa is undoubtedly complex, multifactorial, and not yet completely understood.

#### 2.3.6. Protective Factors

Interestingly, among all the aforementioned pathological molecules, MCs also produce defensive and repairing factors, which altogether alleviate the clinical manifestation in asthmatic patients; additionally, the same protein may play both a damaging and a healing role [60].

Proteoglycans are released along with histamine via degranulation and possess anti-inflammatory properties, reducing NF-κB and antagonizing the action of histamine itself [114]. PGE2 has an anti-inflammatory effect as well, suppressing mast cell degranulation and, therefore, histamine release, and downregulating the production of IL-5 and IL-13 acting on ILC2 [115,116]. Among MCs’ proteases, although chymase and tryptase are infamous for possessing a potent pro-inflammatory activity, tryptase also induces tissue repair stimulating the bronchial epithelium to produce amphiregulin and IL-12B and may cleave several pro-inflammatory cytokines and chemokines including eotaxin 1/3, chemokine (C-C motif) ligand 7 (CCL7), and IL-21 [117,118,119]. Finally, growth factors produced by MCs intrinsically possess an ambivalent function: those are fundamental to healing injured airway tissue, but excessive growth and neovascularization lead inevitably to obstruction and fibrosis [120]. Nevertheless, VEGF-C, which induces lymphangiogenesis, is thought to play a crucial protective role as a decrease in lymphatic vessels was observed in biopsies with fatal asthma, suggesting that lymphatic drainage of mucosal edema from the asthmatic airways may be a crucial factor protecting against death [121].

#### 2.3.7. Non-Atopic Asthma

Patients non-sensitized to common aeroallergens are often described as affected by an “intrinsic” or “non-atopic” asthma. This form is usually more persistent and severe, although the pattern of airway inflammation is virtually the same [122]. As a matter of fact, MCs are increased in number in the bronchial mucosa of both atopic and nonatopic asthmatic subjects compared to healthy controls [123].

In occupational asthma (a particular form of asthma developed or exacerbated following specific exposure in the workplace) the role of MCs is proven as well: for example, in toluene diisocyanate (TDI) asthma, MCs are elevated in the bronchial epithelium and electron microscopy shows that the majority of MCs are degranulated [124]. Thus, MCs appear to contribute to the pathophysiology of occupational asthma similarly to atopic and nonatopic asthma.

Finally, in exercise-induced asthma (a form of asthma where the bronchoconstriction happens after physical exercise) the role of MCs can be related to the increased concentrations of circulating histamine in the serum of asthmatic subjects following exercise [125] and the increased concentrations of histamine, cysteinyl leukotrienes, and tryptase in the sputum following exercise [126].

**Table 1 ijms-24-09771-t001:** Main MC mediators in asthma, focusing on biological and clinical effects. The information contained in the table has been obtained and summarized from the article by Peter Bradding, Andrew F. Walls, and Stephen T. Holgate [38].

Mediator	Function	Clinical Effect
**Histamine**	Bronchoconstriction Mucosal edema Airway remodeling	Bronchospasm Airflow limitation Irreversible obstruction
**Heparin**	Reduce mast cells’ production of pro-inflammatory cytokines	Immunomodulant
**Chymase**	Airway remodeling	Irreversible obstruction
**Tryptase**	Bronchoconstriction	Airflow limitation
**Serotonin**	ASM proliferation	Airflow limitation
**Il-4**	Immune cell recruitment Airway remodeling	Prolong inflammation and its effects
**IL-6**	MCs’ proliferation and activation	Prolong inflammation and its effects
**IL-13**	Mucus production tissue repair Increasing histamine signaling	Airflow limitation Irreversible obstruction
**TNF-** **α**	Production of pro-inflammatory cytokines Proliferation of airway epithelium	Prolong inflammation and its effects Airflow limitation
**Chemokines (e.g., CCL1, CCL2, CCL3, CCL4, CCL5, CCL7)**	Immune cell recruitment	Prolong inflammation and its effects
**Growth factors (e.g., VEGF-A, VEGF-C)**	Lympho-angiogenesis Airway remodeling	Irreversible obstruction

#### 2.3.8. Mast-Cell Centered Therapies in Asthma

MCs play a critical role in asthma and, unsurprisingly, most asthma therapies have a significant effect on these cells. In theory, there would be numerous strategies to inhibit MC activity; for example, it would be possible to target specific MC mediators, block receptors for MC mediators, inhibit MC activation, limit MC expansion, and finally, induce MC apoptosis [127]. To date, the repertoire of asthma therapies does not exploit all of the aforementioned possibilities, but numerous novel anti-MC therapies currently are under investigation. The goal of asthma management is to obtain a good symptom control while reducing the risk of asthma-related mortality, exacerbations, airflow obstruction, and treatment-related side effects; this might require a combination of pharmacological and non-pharmacological measures that should be tailored and personalized according to individual patient characteristics [128]. Since a detailed description of asthma management goes beyond the scope of this review, this paragraph will only provide an overview of the effects that asthma therapies have on MCs.

Inhaled corticosteroids (ICS) and bronchodilators are the mainstay of asthma therapy. Despite bronchodilators mainly acting at the smooth muscle level, several studies have shown that β-agonists induce mast cell stabilization via MC β2-adrenoreceptors [129]. β-agonists have been shown to display different degrees of inhibition of IgE-mediated MC responses, depending on their potency and efficacy, but the high degree of tolerance to these compounds known to occur in smooth muscle cells develops even more rapidly in masts cells [129]. As previously elucidated, corticosteroids act at the transcriptional level, suppressing numerous stages of the inflammatory response and ultimately resulting in the inhibition of pro-inflammatory pathways in immune cells, including MCs [130].

Another strategy is LTRAs, which block signaling through leukotrienes, proinflammatory mediators that in asthmatic patients have been proven to contribute to bronchoconstriction, mucus secretion, airway edema, and inflammatory cell recruitment [131]. MCs are a relevant source of leukotrienes, and they might be activated by leukotrienes as well [131]. LTRAs are generally delivered orally, inhibiting type 2 inflammation throughout the entire respiratory tract, which might be particularly relevant for patients with comorbid upper and lower airway diseases [132]. Nevertheless, when used as monotherapies, LTRAs have been proven to have a lower efficacy compared to ICS [133].

In the past two decades, several biologics meant to selectively inhibit type 2 inflammatory pathways have been approved for the treatment of severe asthma. These include omalizumab, mepolizumab, benralizumab, reslizumab, and dupilumab. The anti-IgE monoclonal antibody omalizumab reduces MC activation, preventing one of the most important triggers for MC degranulation, namely the binding between FcεRIs and IgEs [134]. Unexpectedly, ligelizumab, another anti-IgE biological agent with different IgE-blocking properties, was not found superior to placebo in asthma whereas it is effective in chronic spontaneous urticarial [135,136]. This suggests that distinct IgE monoclonal antibodies may be selectively efficacious in different MCs-mediated diseases. Moreover, even if IgE-mediated MC activation is pivotal in asthmatic individuals with an allergic phenotype, this might not the case for individuals with an eosinophilic-predominant inflammation, where other triggers for MC activation appear to be more relevant [127]. Although IL-5 exerts its main function on eosinophils, MCs possess receptors for this cytokine and MCs may be activated by interaction with eosinophils as well [137]. These findings suggest that also therapies directed against the IL-5 pathway (e.g., mepolizumab, benralizumab, reslizumab) might relevantly influence MC functions. Finally, dupilumab, a monoclonal antibody that blocks the activity of IL-4 and IL-13, besides its other functions, acts indirectly on MCs, as both IL-4 and IL-13 are required for IgE production and are critical mediators of the entire type 2 inflammatory response [138].

Tezepelumab is a new monoclonal antibody recently shown to be effective for the treatment of severe uncontrolled asthma [139]. It acts through the inhibition of thymic stromal lymphopoietin (TSLP), an alarmin mainly derived by epithelial cells, that promotes type 2 inflammation at different stages, including on MCs [140]. Tezepelumab treatment results in a reduction of the eosinophilic infiltrate in the bronchial mucosa; in contrast, this was not observed for other inflammatory cells, including MCs [141]. Yet, patients treated with tezepelumab have a reduced airway hyperresponsiveness and this effect may be due to inhibition of MCs, as TSLP mediates the activation of MCs located in ASMs and airway hyperresponsiveness is generally not improved by biologics that inhibit specifically eosinophilic inflammation [141].

Several promising MC-directed agents are a matter of clinical investigation. For example, monoclonal antibodies that target other cytokines involved in asthma pathophysiology (e.g., the alarmins IL-33 and IL-25) are being studied. Indeed, alarmins are proven to be produced by airway epithelium in response to various stimuli (such as infectious agents, environmental allergens and atmospheric pollutants) and to contribute in asthma pathogenesis and exacerbations: this recent acknowledge has contributed to confirm how epithelium should not be considered only a simple passive barrier, but an immunologically active organ and a promising therapeutical target [142,143,144]. Moreover, biologics that bind MCs and eosinophils’ surface molecules and eventually cause antibody-mediated cytotoxicity of these cells (e.g., lirentalimab, which targets sialic acid-binding Ig-like lectin 8) are under assessment for the treatment of type 2 inflammatory diseases, including asthma [119]. Another field of investigation in MC-mediated disorders are drugs that inhibit KIT activity, resulting in MC apoptosis and depletion, which can be achieved through monoclonal antibodies (e.g., CDX-0159) or small molecules (e.g., imatinib) [145]; the latter was able to decrease airway responsiveness, MC counts, and tryptase release compared to placebo in a proof-of principle trial on severe asthma patients [146]. Another interesting approach would be to inhibit intracellular transduction pathways that mediate immune cell activation, including of MCs, for example, through the blockade kinases (e.g., Janus kinase, Bruton’s tyrosine kinase, spleen tyrosine kinase). Some of the small molecules that inhibit these kinases, already in use for other hematological and immunological disorders, are under investigation also in asthma. For example, a proof of activity trial demonstrated that the topical Janus kinase inhibitor GDC-0214 reduces the fraction of exhaled nitric oxide in patients with mild asthma [147], and other clinical trials on JAK inhibitors in asthma are ongoing [147].

To conclude, the potential targets to inhibit MCs in asthma and other MC-mediated disorders are numerous and it is likely that in near future novel therapies will be available. A better understanding of MCs’ biology and complex phenotypic heterogenicity is critical to provide further insights to drive further research in drug development [145].

## Figures and Tables

**Figure 1 ijms-24-09771-f001:**
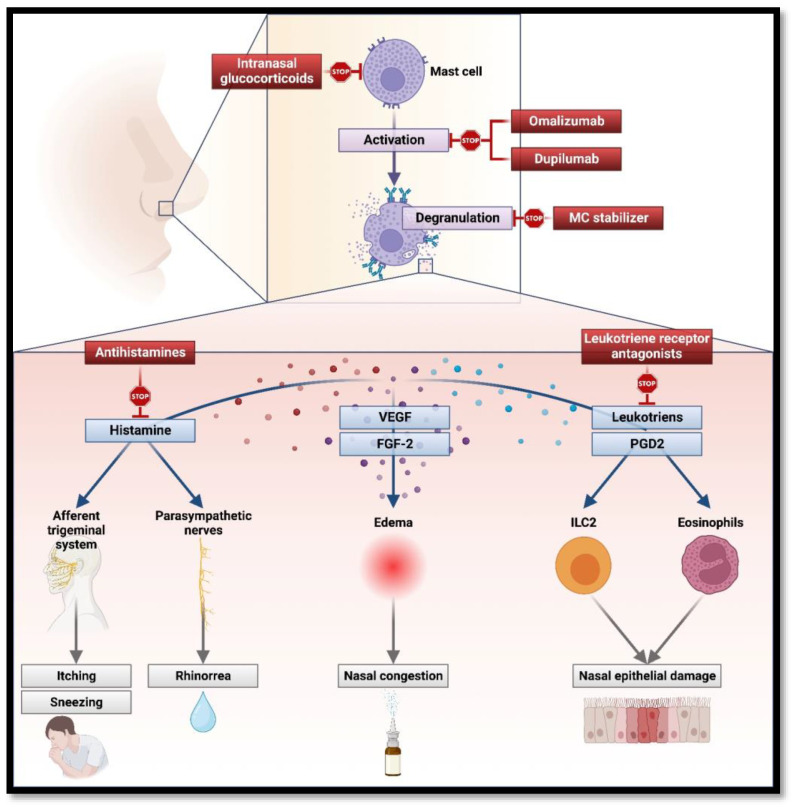
Main mast cell mediators, their role in the pathogenesis of allergic rhinitis and interaction with drugs (created with BioRender.com).

**Figure 2 ijms-24-09771-f002:**
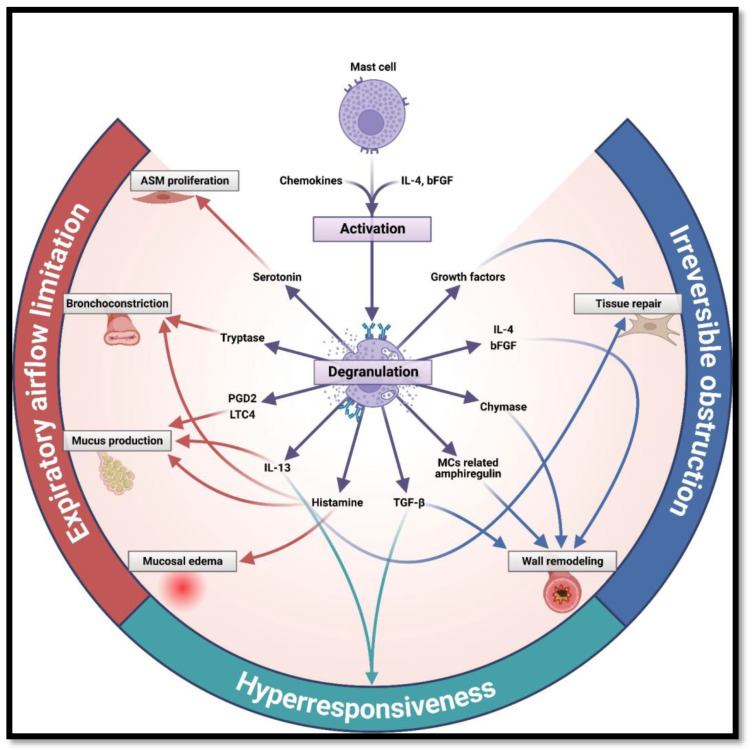
Main mast cell mediators and their role in asthma’s pathogenesis (created with BioRender.com).

## Data Availability

Not applicable.

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
