# Peer review of "Mast Cells in Upper and Lower Airway Diseases: Sentinels in the Front Line"

_ijms, 2023, doi:10.3390/ijms24119771_

Round 1

Reviewer 1 Report

This is a very comprehensive review by Costanzo G et al with focus on mast cell biology and mast cell contribution to lower and upper airway disease. I only have minor comments that may strengthen this review:

1-Table 1: I would position this table later in the review as it seems to mainly list mediators associated with upper and lower airway disease. It is also unclear whether the list was created and or reproduced from ref. 38. If that's the case, I would cite the reference without showing the list. 

2-As the authors describe mast cell role in homeostasis vs. disease, it may be of interest for readers to understand better the potential mechanisms that may lead to dysregulation of mast cell function and their contributions to disease.  For example, contributions of epithelium-derived cytokines (IL-33, TSLP, IL-25) to mast cell pro-inflammatory phenotypes in airway disease. This will also assist the readers to understand the importance of therapies targeting these epithelium-derived cytokines, described later in the review. 

Author Response

We would like to deeply thank the Reviewer’s precious comments and consideration.

As wisely suggested, we proceeded to reallocate Table 1 giving further details about the references used to fill it. We also expanded the paragraph about the role of alarmins in asthma pathogenesis, providing informations useful to the readers to better understand the rationale behind a therapeutical approach. As for the mindful comment about the mechanisms of mast cells dysregulation, it is an interesting yet mostly unknown field of study: We proceeded to provide a specification about the potential role of this topic.

Reviewer 2 Report

1. This paper is an excellent review focusing on the mast-cells (MC) attributes and a comprehensive description of their function in the complex pathophysiology of upper and lower airway disease.

2. The review is extremely clear, concise, comprehensive and relevant for the field of inflammatory diseases of the upper and lower airways. The paper covers a significant gap in the knowledge of the biology and function of MC. 

3.  Although there are studies published in the literature that cover a single specific and singular characteristic and function of MC in inflammatory diseases of the respiratory tract, this review paper is the first complete synthesis about MC and practical ways       of treatment. 

4.  No improvements are to be performed on this excellent synthesis. 

5.  The conclusions presented are coherent and supported by the references listed. 

6.  The reference list is heavily updated and no relevant paper is omitted.

7.  The figures and tables included are appropriate, clearly describing the data and easy to understand.

The paper is a must for those interested in the biology and treatment of inflammatory diseases of the upper and lower airways.

Author Response

We would like to deeply thank the Reviewer’s precious comments and consideration. In particular, we are glad that the Reviewer has appreciated the approach chosen for the article and the effort to summarize such a complex yet interesting subject.

Reviewer 3 Report

In this review, the authors start with a brief overview on mast cells, then focus on the role of mast cells in upper and lower airway diseases.

In the receptors section (1.3) the authors should discuss different families of Fc receptors on mast cells.

The authors should also discuss the importance of MRGPRX2 and Neupeptide.

In the signal transduction section (1.4) the authors should discuss calcium signaling.

In the granules section (1.5) the authors should discuss different methods of staining for mast cells.

In Table 1, the authors should list references.

2.1 Change NAS to NAR line 274.

In addition to skin testing and IgE testing, the authors should mention antigen challenge/food challenge as a diagnosis (line 300).

In the Omalizumab section line 384, the authors should mention how the drug works: it binds to the Fc portion of IgE, preventing IgE from binding to its receptor, though IgE remains in circulation this leads to a reduction in FceR1 on the cell surface (specifically on B cells and basophils).

The authors should describe immunotherapy a bit more (line 402). No description of the process is given.

The authors should fix the reference at line 461.

at line 722, the authors should include TSLP as an alarming.

Author Response

We would like to deeply thank the Reviewer’s precious comments, addressing the challenges of summarizing such a complex yet interesting subject. Before answering every comment, w would like to specify that all the Authors of this manuscript agreed to deliberately focus the brief initial overview on the most essential information to better understand the following paragraphs.

REVIEWER'S COMMENT: In the receptors section (1.3) the authors should discuss different families of Fc receptors on mast cells.

RESPONSE: We better specified the presence of FcγRI in the list of the various MC receptors, although the role in the pathogenesis of the airway diseases is yet to be fully understood.

REVIEWER'S COMMENT: The authors should also discuss the importance of MRGPRX2 and Neupeptide.

RESPONSE: We provided some brief information on MRGPRX2 and Neuropeptide receptors in the revised version of the article.

REVIEWER'S COMMENT: In the signal transduction section (1.4) the authors should discuss calcium signaling.

RESPONSE: We cited the calcium-release mechanism among the signaling mechanisms, in the revised manuscript.

REVIEWER'S COMMENT: In the granules section (1.5) the authors should discuss different methods of staining for mast cells.

RESPONSE: We added a brief indication on MCs staining in the revised manuscript

REVIEWER'S COMMENT: In Table 1, the authors should list references.

RESPONSE: We added references details for Table 1 in its legend.

REVIEWER'S COMMENT: 2.1 Change NAS to NAR line 274.

RESPONSE: We corrected NAS to NAR, as suggested.

REVIEWER'S COMMENT: in addition to skin testing and IgE testing, the authors should mention antigen challenge/food challenge as a diagnosis (line 300).

RESPONSE: We thank the Reviewer for this comment. We added a comment on nasal allergen challenge. We did not added food challenges as they are not part of the diagnostic algorithm of respiratory allergic diseases.

REVIEWER'S COMMENT: In the Omalizumab section line 384, the authors should mention how the drug works: it binds to the Fc portion of IgE, preventing IgE from binding to its receptor, though IgE remains in circulation this leads to a reduction in FceR1 on the cell surface (specifically on B cells and basophils).

RESPONSE: We provided further information about the therapeutical mechanism of Omalizumab 

REVIEWER'S COMMENT: The authors should describe immunotherapy a bit more (line 402). No description of the process is given.

RESPONSE: We provided further information about the therapeutical mechanism of allergen immunotherapy

REVIEWER'S COMMENT: The authors should fix the reference at line 461.

RESPONSE: Thanks for this comment. We already corrected this typo in the revised version already submitted after Reviewers 1 and 2 comments. 

REVIEWER'S COMMENT:  at line 722, the authors should include TSLP as an alarming.

 RESPONSE: We proceed to mention TSLP as an alarmin in the revised manuscript.
